# Coordinated genomic control of ciliogenesis and cell movement by RFX2

**Mei-I Chung[1][†], Taejoon Kwon[1][†], Fan Tu[1], Eric R Brooks[1], Rakhi Gupta[2], Matthew Meyer[1], Julie C Baker[2], Edward M Marcotte[1,3,4], John B Wallingford[1,3,4,5]***

[1]Department of Molecular Biosciences, University of Texas at Austin, Austin, United States; [2]Department of Genetics, Stanford University, Stanford, United States; [3]Center for Systems and Synthetic Biology, University of Texas at Austin, Austin, United States; [4]Institute for Cellular and Molecular Biology, University of Texas at Austin, Austin, United States; [5]Howard Hughes Medical Institute, University of Texas at Austin, Austin, United States

**Abstract** The mechanisms linking systems-level programs of gene expression to discrete cell biological processes in vivo remain poorly understood. In this study, we have defined such a program for multi-ciliated epithelial cells (MCCs), a cell type critical for proper development and homeostasis of the airway, brain and reproductive tracts. Starting from genomic analysis of the cilia-associated transcription factor Rfx2, we used bioinformatics and in vivo cell biological approaches to gain insights into the molecular basis of cilia assembly and function. Moreover, we discovered a previously unrecognized role for an Rfx factor in cell movement, finding that Rfx2 cell-autonomously controls apical surface expansion in nascent MCCs. Thus, Rfx2 coordinates multiple, distinct gene expression programs in MCCs, regulating genes that control cell movement, ciliogenesis, and cilia function. As such, the work serves as a paradigm for understanding genomic control of cell biological processes that span from early cell morphogenetic events to terminally differentiated cellular functions.

*For correspondence:
wallingford@austin.utexas.edu

[†]These authors contributed equally to this work

**Competing interests:** The authors declare that no competing interests exist.

**Reviewing editor**: Marianne E Bronner, California Institute of Technology, United States

## Introduction

A major goal of biology over the last several decades has been to understand the mechanisms that control differential gene expression. While recent advances in genomic technology have dramatically empowered these studies, we still know comparatively little about the mechanisms linking systems-level programs of gene expression to discrete cell biological processes in vivo. This gap in our understanding is important because organ and tissue function are ultimately executed by the specialized behaviors of individual cells (e.g., polarized secretion in excretory organs, coordinated contraction in muscle cells).

Recent studies of cilia and ciliated epithelial cells highlight the current disconnect between genomics and cell biology: it is clear that hundreds of proteins are required for cilia assembly and function (*Gherman et al., 2006*; *Inglis et al., 2006*), and moreover, assembly of new cilia/flagella clearly requires new transcription (*Thomas et al., 2010*). Nonetheless, only a handful of transcription factors have been identified that control cilia structure and function, and their associated gene regulatory networks remain largely undefined, especially in vertebrates. In *C. elegans*, the sole RFX family transcription factor *daf-19* is the central regulator of ciliogenesis, and dozens of target genes are known to effect its action (*Efimenko et al., 2005*; *Phirke et al., 2011*; *Burghoorn et al., 2012*). The one Rfx factor in *Drosophila* is likewise well characterized (*Laurencon et al., 2007*; *Newton et al., 2012*). By contrast, multiple RFX family members are essential for ciliogenesis in vertebrates, but as yet, there has been no comprehensive genome-wide survey of Rfx-dependent gene expression as it relates to ciliogenesis (*Bonnafe et al., 2004*; *Ashique et al., 2009*; *El Zein et al., 2009*). This gap in our knowledge of the genomics of RFX factors is made the more important because these proteins also possess cilia-independent functions about which very little is

**eLife digest** Cells that have hundreds of tiny hair-like structures called cilia on their surface have important roles in our airways and also in the brain and reproductive system. By beating in a coordinated manner, the cilia cause fluid to flow in a particular direction. The development of these multiciliated cells is a complex process in which genes are expressed as proteins, with this gene expression being regulated by other proteins called transcription factors.

In invertebrates the development of the cilia is controlled by transcription factors from the RFX family, which also appear to be important for development of cilia in vertebrates. However, the details of this process—in particular, the identities of the genes that are involved and how their functions are related—are not well understood in vertebrates.

Chung et al. have sought to remedy this by analyzing the network of genes whose expression is controlled by the transcription factor Rfx2 in vertebrates. The results showed that the genes controlled by Rfx2 were involved in all aspects of cilia, including several genes that are known to be mutated in diseases caused by abnormal cilia. Chung et al. also identified genes that were not previously thought to be relevant to cilia.

As multiciliated cells are developing, but before they can generate cilia, they must first migrate from the bottom of the epithelium, the layer of tissue in which they function, to the top of this layer. Chung et al. found that Rfx2 was also involved in this process.

The approach taken by Chung et al.—which involved a combination of RNA sequence analysis, examination of Rfx2 binding sites on chromosomes, computational predictions of protein interactions and in vivo cellular imaging—could be used to perform similar systems-level analyses of other developmental and biological processes.

yet known, including control neuronal and pancreatic development (e.g., *Senti and Swoboda, 2008*; *Ait-Lounis et al., 2010*; *Pearl et al., 2011*; *Benadiba et al., 2012*).

One vertebrate cell type in which Rfx factors are known to play particularly important roles is the multi-ciliated epithelial cell (MCC). These cells project dozens or hundreds of motile cilia from their apical surfaces, and the polarized beating of these cilia generates fluid flow that is essential for development and homeostasis in many organ systems (*Figure 1*). Such cells are central to the normal homeostasis of airway, brain and reproductive tracts (*Worthington and Cathcart, 1963*; *Yeung et al., 1991*; *Lyons et al., 2006*; *Fahy and Dickey, 2010*), and defective functioning of these cells is associated with pathologies ranging from chronic infection to hydrocephalus (*Afzelius, 1976*). Despite these cells' importance and long history of study (*Sharpey, 1830*), the transcription factors that control MCC development and function are only now being elucidated (*You et al., 2004*; *Stubbs et al., 2008*; *Marcet et al., 2011*; *Stubbs et al., 2012*; *Tan et al., 2013*), and among these factors is Rfx2 (*Chung et al., 2012*).

In this study, we sought to combine systems biology approaches with in vivo cell biological studies in order to better define the genomic control of MCC development and function. We used high-throughput sequencing of Rfx2-regulated transcripts, systematic mapping of Rfx2 chromosomal binding sites, and bioinformatic exploration of functional protein interactions to guide our mechanistic cell biology experiments. This approach provided diverse new insights into the roles played by Rfx2 in ciliogenesis and polarized ciliary beating. Surprisingly, we also find that Rfx2 plays a central, but unanticipated, role in controlling cell movement in newborn MCCs, a process about which almost nothing is currently known. Overall, Rfx2 activates a complex program of gene expression that serves to coordinate several distinct features of MCCs, including cell migration, ciliogenesis, and cilia function, thus serving as a paradigm for genomic control of cell biological processes that span from early differentiation events to terminally differentiated cell functions.

## Results

### Systems-level analysis of Rfx2 function in a vertebrate mucociliary epithelium

To explore the genomic control of cell behavior in MCCs, we turned to the amphibian embryo, which has emerged as a powerful and rapidly assayable in vivo model for this cell type. Indeed, several foundational

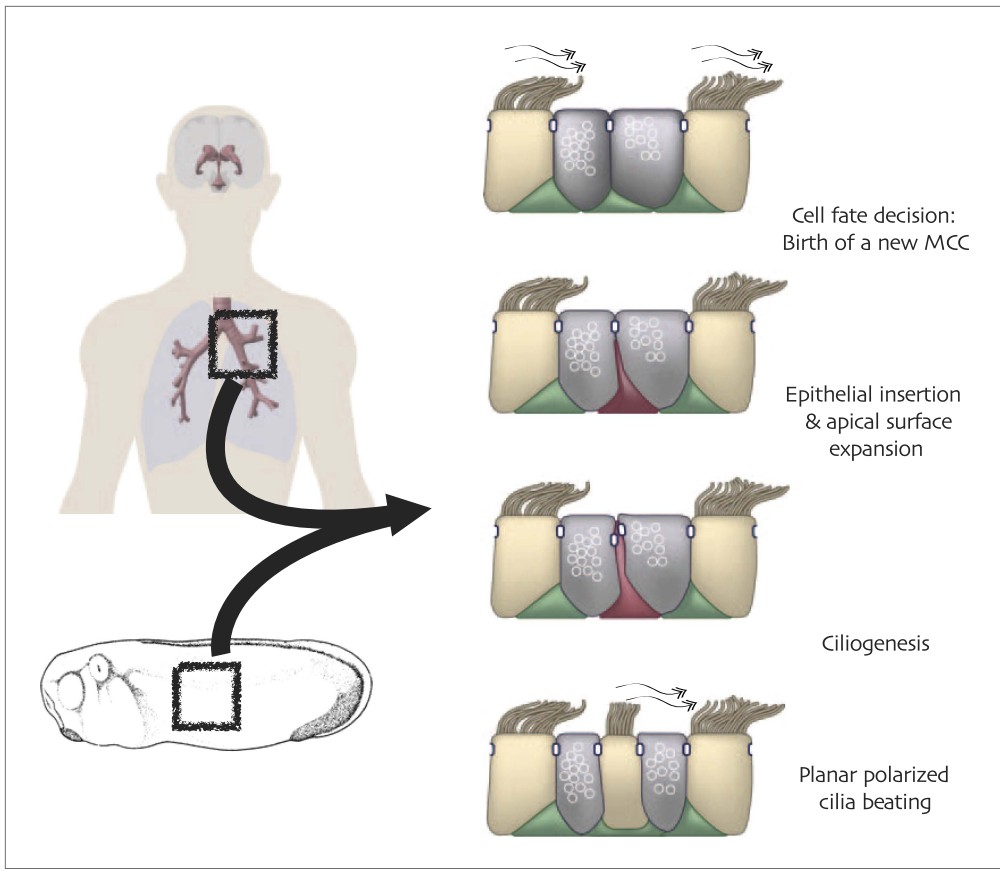

**Figure 1**. Conserved cell behaviors during multi-ciliated cell development in mammalian airways and Xenopus epidermis.

studies of specification, ciliogenesis, and planar polarization in MCCs have been performed in amphibians (*Deblandre et al., 1999*; *Stubbs et al., 2006*; *Park et al., 2008*; *Mitchell et al., 2009*; *Marcet et al., 2011*; *Sirour et al., 2011*; *Werner and Mitchell, 2011*; *Stubbs et al., 2012*). Importantly, these studies have consistently prefigured results in mammals (*Marcet et al., 2011*; *Morimoto et al., 2010*; *Stubbs et al., 2012*; *Tsao et al., 2009*; *Vladar et al., 2012*). We therefore exploited the *Xenopus* system to perform parallel RNA transcript sequencing (RNA-seq) and chromatin immunoprecipitation deep sequencing (ChIP-seq) for Rfx2, which we showed previously to be essential for the normal development of cilia in MCCs (*Chung et al., 2012*) (*Figure 2*).

A detailed description of this approach is provided in the 'Materials and methods' section. Briefly, RNA-seq was performed on isolates of *Xenopus* mucociliary epithelium grown in organotypic culture, comparing control samples to knockdown using validated Rfx2 morpholino antisense oligonucleotides (*Figure 2A*; *Figure 2—figure supplement 1*) (*Chung et al., 2012*). We performed two biological replicates, each containing 100–200 individual tissue isolates; analysis of the correlation found excellent reproducibility between the two replicates (*Figure 2—figure supplement 2*). As we are most concerned here with results that will be relevant to human health, we confined our analysis of the RNA-seq data to *Xenopus* genes for which there were unambiguous human orthologues to facilitate cross-organism comparison. From this list of 11,644 genes, we identified 2750 genes that were differentially expressed following Rfx2 knockdown (>twofold; FDR<0.05) (*Figure 2A*, *Figure 2—figure supplement 2*).

We next used ChIP-seq to ask which of these differentially expressed genes were direct targets of control by Rfx2. We identified 3465 genes that were significantly bound by Rfx2 and, importantly, this set of genes overlapped significantly with the set of genes differentially expressed in Rfx2 morphants (*Figure 2A*, *Figure 2—figure supplement 3*) ($p \leq 10^{-6}$, hypergeometric test). The intersection of these two experiments defined 911 genes (*Figure 2—source data 1*), corresponding to the directly regulated downstream target genes of Rfx2. We tested a subset of these directly bound, differentially expressed

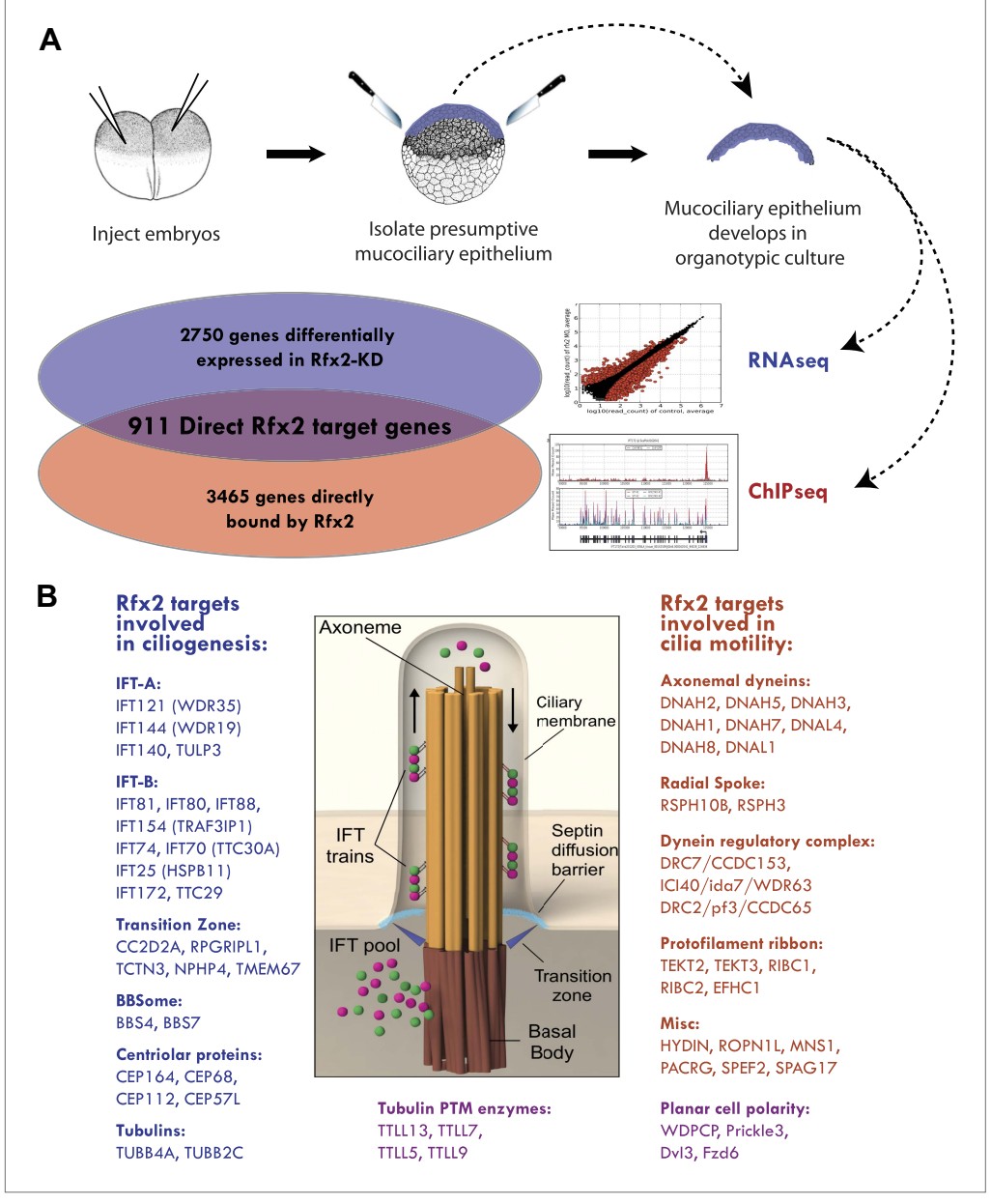

**Figure 2**. Rfx2 controls diverse ciliogenic machinery. (**A**) Schematic diagram of this study. (**B**) Prominent cilia-related genes identified as Rfx2 targets in this study.

The following source data and figure supplements are available for figure 2:

**Source data 1**. 911 genes corresponding to the directly regulated downstream target genes of Rfx2.

**Source data 2**. Table of enriched GO terms.

**Figure supplement 1**. Controls for the morpholino antisense oligonucleotides used in this study.

**Figure supplement 2**. Summary of RNA-seq data.

**Figure supplement 3**. Three examples of RNA-seq and ChIP-seq data.

**Figure supplement 4**. Validation of Rfx2-dependent genes.

genes by in situ hybridization and RT-PCR analysis; 100% were confirmed as Rfx2-dependent genes (*Figure 2—figure supplement 4*). We then focused our further studies on this set of 911 direct targets.

## Rfx2 controls assembly of diverse ciliogenic machinery

An unbiased analysis of enriched Gene Ontology (GO) terms suggested a key role for Rfx2 in the control of ciliary gene expression (*Figure 2—source data 2*), consistent with the known roles for Rfx2 in ciliogenesis (*Chung et al., 2012*). Moreover, direct examination of the 911 Rfx2 target genes revealed that components of essentially all known ciliary machinery are under the control of Rfx2 in MCCs (*Figure 2B*). Principal among these is the intraflagellar transport (IFT) system, the core mechanism for moving cargoes into and out of the cilium. IFT involves two separable complexes, IFT-A and IFT-B (*Pedersen and Rosenbaum, 2008*), and we identified the IFT–B complex components *ift172* and *ift88* as direct Rfx2 targets, consistent with data from flies, worms, and mice (*Thomas et al., 2010*). Moreover, we also identified several additional Rfx2-regulated components of both IFT-B and IFT-A complexes, as well as the IFT-A adaptor *tulp3* (*Mukhopadhyay et al., 2010*) (*Figure 2B*; *Supplementary file 1A*). Additionally, we found that Rfx2 directly controls expression of genes encoding axonemal dynein subunits, components of the transition zone, and a component of the BBSome, and many of these genes are mutated in human ciliopathies (*Figure 2B*; *Supplementary file 1A*) (*Sharma et al., 2008*; *Reiter et al., 2012*).

Significantly, our analysis identified several ciliary systems that have not previously been associated with RFX factors. For example, Rfx2 controlled expression of tubulins, enzymes involved in tubulin post-translational modification, and microtubule binding proteins, such as *map7* and *spef1/clamp*, which localize to the proximal and distal axonemes, respectively (*Brooks and Wallingford, 2012*) (*Figure 2B*; *Supplementary file 1A*). Rfx2 also controlled the expression of the planar cell polarity (PCP) effector gene *fritz/wdpcp*, which encodes a protein governing assembly of the septin diffusion barrier at the base of cilia (*Kim et al., 2010*). Rfx2 also controls many centriolar genes required for ciliogenesis, including *cep164* (*Graser et al., 2007*). Notably, *wdpcp* and *cep164* are both implicated in human ciliopathies (*Kim et al., 2010*; *Chaki et al., 2012*). This analysis thus provides a comprehensive view of Rfx-related ciliary gene expression in vertebrates.

## Genomic analysis of Rfx2 identifies a novel regulator of intraflagellar transport

These data provide a foundation for understanding Rfx-mediated control of known cilia genes. However, a particularly striking feature of the Rfx2 target genes was the large number whose relationship to cilia (or lack thereof) is not known. Because transcriptionally co-regulated genes frequently share functions (*Eisen et al., 1998*; *Marcotte et al., 1999*), we next asked if our Rfx2 target gene set might provide a jumping-off point for exploring the molecular biology of ciliogenesis, with an aim towards implicating new proteins in this important process.

To focus our search, we used the direct Rfx2 target genes as a seed set to interrogate a probabilistic human gene network (HumanNet) that captures functionally linked genes based on observations in large-scale functional genomics and proteomics datasets (*Lee et al., 2011*). Using 'guilt-by-association' in this network, we identified *ttc29* as a candidate for functional interaction with the IFT machinery (*Figure 3A*). We tested this prediction and found not only that Ttc29-GFP localized to ciliary axonemes, but also that knockdown elicited substantial defects in ciliogenesis (*Figure 3B–D*). To test the HumanNet prediction more directly, we used high-speed in vivo confocal imaging of IFT particle dynamics in *Xenopus* MCCs (*Brooks and Wallingford, 2012*). Strikingly, partial knockdown of Ttc29 resulted in a significant decrease in the mean rate of anterograde IFT (*Figure 3E–G*; *Video 1*), consistent with the bioinformatic linkage of *ttc29* to components of the anterograde IFT-B complex (*ift88* and *ttc30a*; *Figure 3A*). Retrograde IFT movement was not significantly affected by Ttc29 knockdown (*Figure 3H*). Thus, by combining our genomic dataset with functional gene networks, we have not only revealed new links between Rfx2 and ciliogenesis, but we have also identified a specific function for an uncharacterized protein in the regulation of anterograde IFT.

## Genomic analysis of Rfx2 targets elucidates the mechanisms governing cilia beating

To further explore the mechanisms of Rfx2 function, we returned to the list of GO terms enriched in Rfx2 target genes. Of these, the most strongly enriched biological process GO term was 'ciliary or flagellar motility' (*Figure 2—source data 2*). A direct exploration of the Rfx2 targets identified a wide

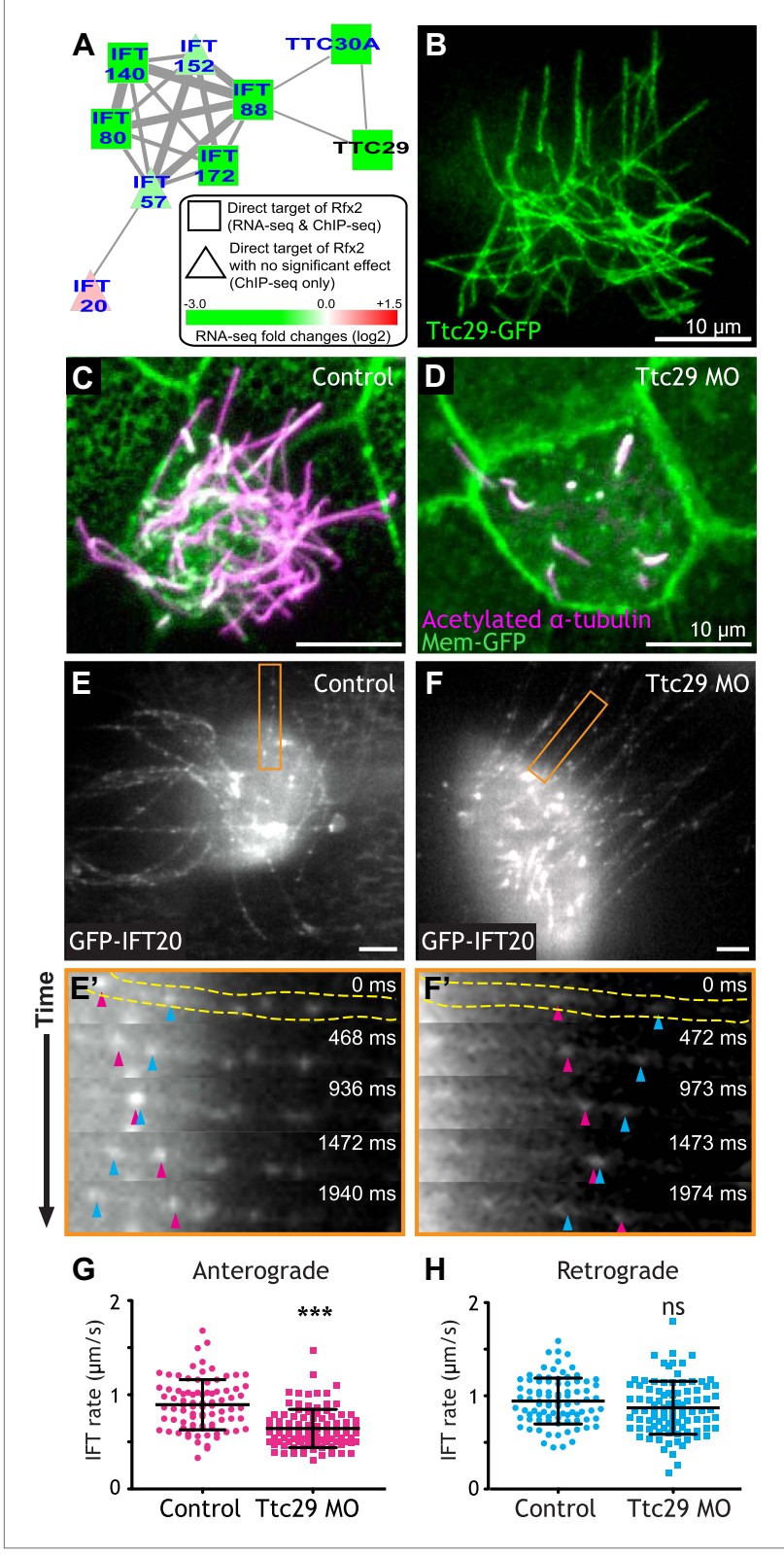

**Figure 3**. Ttc29 is required for ciliogenesis of MCCs by regulating Intraflagellar Transport. (**A**) Ttc29 is clustered with IFT components in HumanNet. (**B**) Ttc29-GFP is localized in the axoneme. (**C**) A MCC of a stage 27 control embryo injected with membrane-GFP. Acetylated α-tubulin labels cilia and GFP labels the cell boundary. (**D**) A

*Figure 3. Continued on next page*

*Figure 3. Continued*

MCC of a stage 27 Ttc29 morpholino-injected embryo. Note that only a few short axonemes are shown following Ttc29 knockdown. (**E**) Still-frame of a control multiciliated cell expressing GFP-IFT20. The axoneme shown in the time series (**E′**) is labeled in orange (***Video 1***). (**E′**) A time-series of a single control axoneme from (**E**) shows processive bi-directional traffic (the distal tip of the axoneme is to the right; pink arrowheads denote an anterograde train over time, blue arrowheads indicate a retrograde train). (**F**) A single still frame from a Ttc29 MO treated multi-ciliated cell expressing GFP-IFT20 (***Video 2***). (**F′**) A time-series of a single axoneme from (**F**). Note that processive bi-directional traffic is qualitatively normal. (**G**) Quantification of anterograde GFP-IFT20 rates shows a significantly slower average anterograde rate upon Ttc29 MO treatment (Control: n = 97 IFT trains, 40 axonemes, 21 Cells, 6 embryos. Ttc29 MO: n = 100 IFT trains, 53 axonemes, 20 cells, 6 embryos. p < 0.0001). (**H**) Quantification of retrograde GFP-IFT20 rates reveals no significant difference between control and Ttc29 MO conditions (Control: n = 87 IFT trains, 40 axonemes, 21 cells, 6 embryos. Ttc29 MO: n = 94 IFT trains, 53 axonemes, 20 cells, 6 embryos. p = 0.0510).

---

range of genes with known roles in cilia beating, including genes implicated in planar polarization of cilia (*frizzled*, *prickle*) (***Mitchell et al., 2009***; ***Vladar et al., 2012***), tubulin polyglutamylases governing inner arm dynein activity (*ttll7*, *ttll9*, *ttll13*), dynein regulatory complex components, radial spoke components, and a variety of poorly-understood cilia beating genes, such as *ropn1l*, *mns1*, *pacrg*, and *hydin* (***Lechtreck et al., 2008***; ***Wilson et al., 2010***; ***Fiedler et al., 2011***; ***Zhou et al., 2012***) (***Figure 2B***; ***Supplementary file 1B***). Given this large array of cilia-beating genes controlled by Rfx2, we turned again to HumanNet, this time with the goal of providing new insights into the molecular control of cilia beating.

Unbiased clustering predicted that three known ciliary beating genes (*dnai1*, *ropn1l*, *mns1*) (***Horvath et al., 2005***; ***Fiedler et al., 2011***; ***Zhou et al., 2012***) should be functionally linked to *ribc2*, which encodes an uncharacterized vertebrate protein with similarity to *Chlamydomonas* protofilament ribbon proteins (***Linck and Norrander, 2003***) (***Figure 4A***). Using a GFP-fusion, we found that Ribc2 localized to axonemes, though knockdown did not substantially affect cilia length (***Figure 4B–D***). Rather, high-speed in vivo imaging of axonemes and analysis of fluid flow revealed a specific defect in ciliary beating following disruption of Ribc2 function (***Figure 4E–H***; ***Videos 3 and 4***). Transmission electron microscopy revealed that loss of Ribc2 did not affect the integrity of outer doublet or central pair microtubules, but did disrupt their organization within the axoneme and the apparent number of dynein arms (***Figure 4I,J***, ***Figure 4—figure supplement 1***).

Little is known about protofilament ribbon proteins in any system (***Linck and Norrander, 2003***), but guided by connections in HumanNet (***Figure 4A***), we found that Ribc2 is essential for the normal axonemal localization of another protofilament ribbon protein, Tekt2 (***Figure 4K–M***). Moreover, our approach also led us to assign a cilia beating function to the previously uncharacterized kinase Nme5 (***Figure 4—figure supplement 2***) and to find that proper localization of this kinase within axonemes is Ribc2-dependent (***Figure 4N***).

These data highlight the power of combining genomic data with functional gene network analysis: The approach identified new roles for Ribc2 and Nme5 in cilia beating and placed these proteins in a functional hierarchy with Tekt2. By contrast, *pacrg* is another direct target of Rfx2 that is required for cilia motility (***Wilson et al., 2010***) but is not linked to *ribc2* in HumanNet, and Ribc2 knockdown did not affect the normal axonemal localization of Pacrg protein (***Figure 4—figure supplement 3***). Thus, our combination of genomics, bioinformatics, and in vivo cell biology provide an effective paradigm for understanding the links between system-level gene expression and discrete behaviors of individual cells.

*CtlEmbC2_crop1 - created with Imaris 7.6*

**Video 1**. Dynamics of GFP-IFT20 in a control multiciliated cell. A control multiciliated cell expressing GFP-IFT20 is shown. Processive bidirectional traffic can be observed. Also see ***Figure 3***.

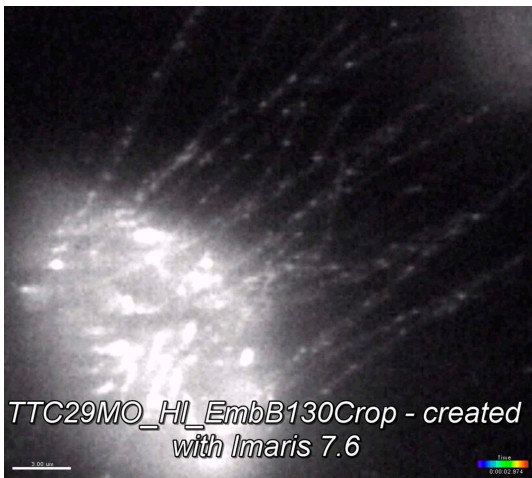

*TTC29MO_HI_EmbB130Crop - created with Imaris 7.6*

**Video 2**. Dynamics of GFP-IFT20 in a Ttc29-knockdown multiciliated cell. A multiciliated cell of a Ttc29-knockdown embryo is shown. Anterograde GFP-IFT20 traffic showed a significantly slower average rate following Ttc29 knockdown. Also see *Figure 3*.

## Rfx2 is essential for the insertion of nascent MCCs into the mucociliary epithelium

Our data demonstrate that Rfx2 governs ciliogenesis and cilia beating via the expression of myriad genes, including many novel genes characterized here for the first time. Interestingly however, of the 911 Rfx2 target genes, only ~20% are present in known cilia proteomes (*Figure 5A*). This finding suggests that many of the Rfx2 target genes are NOT involved in ciliogenesis or cilia function, a notion that is intriguing in light of recent reports of cilia-independent roles for 'ciliary' RFX genes (e.g., *Senti and Swoboda, 2008*; *Ait-Lounis et al., 2010*).

Analysis of GO terms associated with Rfx2 target genes suggested a potential role for Rfx2 in cell movement and cell morphogenesis (*Figure 5A*, *Figure 2—source data 2*), which is of interest because both *Xenopus* MCCs and their counterparts in mammalian airways arise from a population of p63-expressing basal precursor cells (*Figure 1*, red cell at right) (*Evans and Moller, 1991*; *Drysdale and Elinson, 1992*; *Lu et al., 2001*; *Rock et al., 2010*). Consistent with a role for Rfx2 in the apical movement of MCCs, we found that strong knockdowns consistently resulted in MCCs being positioned well below the apical surface of the epithelium (*Figure 5B,C*). This effect was not secondary to a ciliogenesis defect, as milder knockdowns with lower doses of MO did not inhibit MCC emergence but did suppress ciliogenesis (*Chung et al., 2012*). These results suggested a previously unrecognized role for an Rfx factor in cell movement.

A role for Rfx2 in apical movement is noteworthy, because this process is a conserved feature of newly born MCCs (*Evans et al., 1989*; *Drysdale and Elinson, 1992*; *Lu et al., 2001*; *Rock et al., 2009*), and it must involve apical movement, remodeling of junctions, and assembly of an apical cell surface to which basal bodies can dock prior to ciliogenesis. The mechanisms guiding these crucial aspects of MCC biology are largely unknown, so we used transgenic drivers to direct expression of fluorescent reporters specifically in MCCs and documented these cells' behavior using 4D confocal imaging in vivo.

We identified two broad categories of MCC behavior associated with apical insertion. MCCs first engaged in a probing of neighboring cell–cell boundaries; subsequently, this behavior ceased and MCCs smoothly expanded their apical surfaces (*Figure 5D*; *Videos 5 and 6*). Time-lapse imaging in Rfx2 morphants revealed that the initial probing behavior was qualitatively normal, while the latter phase of apical surface expansion failed completely (*Figure 5E*; *Video 7*).

## Rfx2 acts cell-autonomously to control apical surface expansion in nascent MCCs

Rfx2 is expressed strongly in MCCs (*Chung et al., 2012*) but RT-PCR also detected expression of Rfx2 in the neighboring goblet cells (not shown). Thus, Rfx2 might act in MCCs to control their cell movement and insertion into the epithelium or it may control behaviors in neighboring superficial cells that facilitate the process—or both. To distinguish between these possibilities, we generated mosaics in which Rfx2 was disrupted specifically in MCCs or specifically in the overlying goblet cells. Knockdown of Rfx2 in the MCCs led to a robust failure of MCC apical surface expansion while knockdown in overlying superficial cells had no effect (*Figure 6*). Thus our data suggest that Rfx2 cell-autonomously controls the process of MCC apical surface expansion.

## Rfx2 controls insertion of nascent MCCs via *dystroglycan* and *slit2*

Insertion of MCCs into the mucociliary epithelium represents a novel cell behavior, the molecular mechanisms of which remain almost entirely obscure. Given the specific role for Rfx2 in this process, we leveraged our genomic dataset to gain insights. Consistent with the observed insertion defects, we found

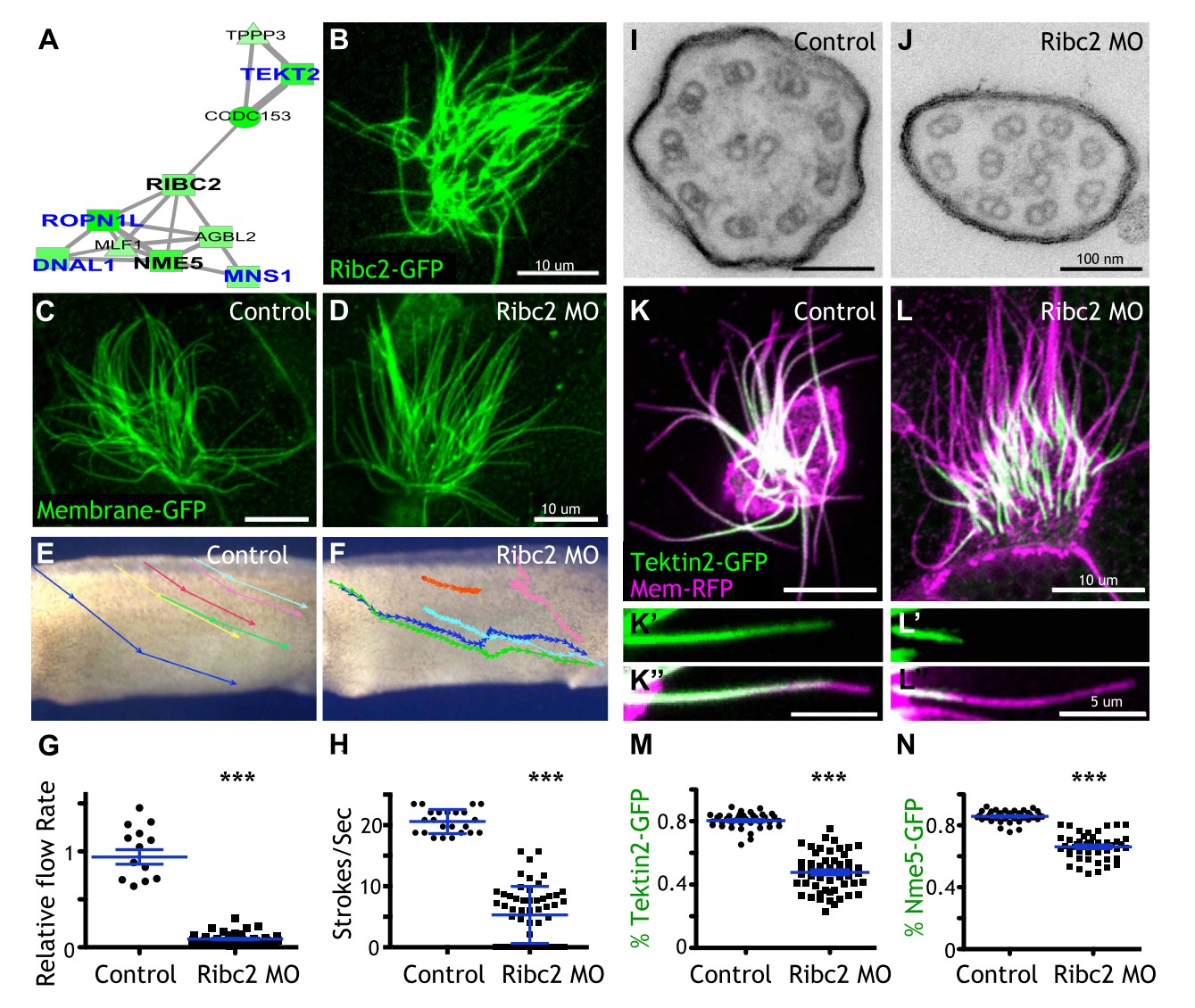

**Figure 4**. Ribc2 is required for ciliary motility. (**A**) Ribc2 is clustered in HumanNet with known ciliary beating components, such as Dnal1, Ropn1l, and Mns1 (**B**) Ribc2-GFP is localized along the axoneme. (**C**) An MCC of a stage 27 control embryo injected with membrane-GFP. (**D**) An MCC of a stage 27 embryo injected with Ribc2 morpholino. Ribc2 is not essential for cilia assembly. Tracking of latex beads moving across the epidermis of the control embryo (**E**) and the Ribc2 morphant (**F**). An arrow represents the moving distance per time frame. The relative average flow rate is shown in (**G**). While the average flow rate of control is normalized to 1 ± 0.075 (mean ±SEM), it is significantly reduced to 0.085 ± 0.008 in Ribc2 morphants. (**H**) Quantification of ciliary beating using high-speed confocal (**Videos 3 and 4**). Beat frequency is 20.59 ± 0.410 strokes/s in control whereas only 5.29 ± 0.635 strokes/s following Ribc2 knockdown. Ultrastructure of axoneme from a control embryo (**I**) and a Ribc2 knockdown embryo (**J**) were visualized using TEM. Lack of dynein arms were observed in Ribc2 morphants. (**K**) A MCC of a stage 27 control embryo injected with Tektin2-GFP and membrane-RFP. Enlarged view of an axoneme is shown in (**K′**) (**K″**). (**L**) A MCC of a stage 27 Ribc2 morphant. Enlarged view is shown in (**L′**) (**L″**). (**M**) Tektin2-GFP generally decorates 80% (±0.8) of the axoneme as marked by membrane-RFP; this ratio is significantly reduced, to 48% (±1.6), following Ribc2 knockdown (**N**). Nme5-GFP generally decorates 86% (±0.5) of the axoneme; this ratio is significantly reduced, to 66% (±1.2), following Ribc2 knockdown. ***$p < 0.0001$ Mann–Whitney test.

The following figure supplements are available for figure 4:

**Figure supplement 1**. Ribc2 is required for axonemal organization.

**Figure supplement 2**. Nme5 is required for ciliary motility.

**Figure supplement 3**. Ribc2 is not required for the axonemal localization of Pacrg-GFP.

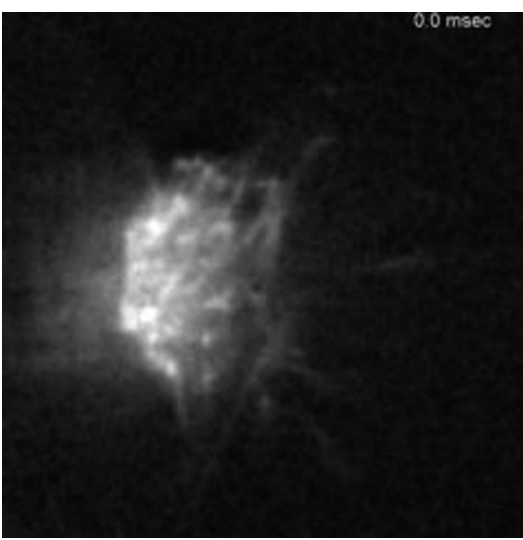

**Video 3**. Cilia beating of a control multiciliated cell. A control multiciliated cell expressing membrane-GFP is shown. Beat frequency is 20.59 ± 0.410 strokes/s in control multiciliated cells.

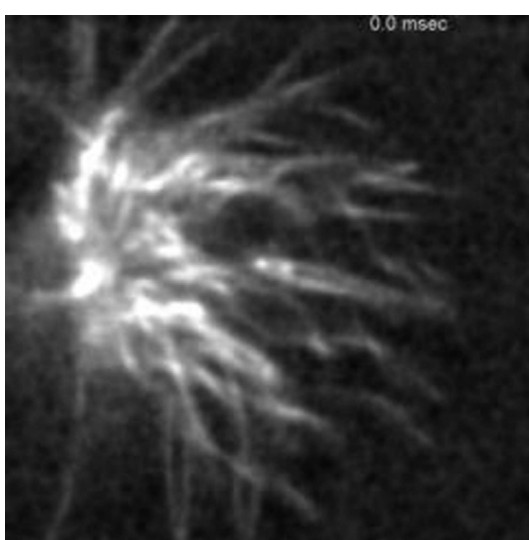

**Video 4**. Cilia beating of a Ribc2-knockdown multiciliated cell. A Ribc2-knockdown multiciliated cell expressing membrane-GFP is shown. Beat frequency is 5.29 ± 0.635 strokes/s following Ribc2 knockdown.

that Rfx2 directly regulates the extracellular matrix component *dystroglycan* (*dag1*), which is a known regulator of MCC insertion (*Sirour et al., 2011*). Interestingly, Dag1 is known to functionally interact with the Slit/Robo signaling system to control cellular morphogenesis in diverse settings (*Medioni et al., 2008*; *Wright et al., 2012*), so it was notable that Rfx2 directly controlled transcription of both the *slit2* ligand and the intracellular effector *srgap2* (*Supplementary file 1C*). Accordingly, we found that Slit2 knockdown disrupted insertion of nascent MCCs into the mucociliary epithelium and analysis of mosaic embryos revealed that Slit2 acts cell-autonomously in MCC insertion, as was the case for Rfx2 (*Figure 7*). Together, these data identify *rfx2*, *dag1*, and *slit2* as a preliminary molecular framework for epithelial insertion of nascent MCCs and reveal a tight genomic coordination of cell movement, ciliogenesis, and cilia motility.

## Discussion

During development, many cell types must first execute a specific set of transient behaviors before finally differentiating into a final, functional form. For example, many cells must first undergo transitions from mesenchyme to epithelium or vice-versa, must engage in migration, or must radically change their shape. Then, as they terminally differentiate, most cells will assemble specialized cellular machinery in order to execute their cell-type specific functions. Understanding how disparate cell types execute specialized cell behaviors is therefore a key challenge in developmental biology and will require us not only to delineate gene regulatory networks, but also to associate particular genes in the network with particular cellular machinery. Meeting this challenge will require the application of genomic technologies specifically to cell biological questions in vivo.

In this study, we explored the role of the Rfx2 transcription factor in MCCs, an essential cell type for homeostasis of the brain, airway, and reproductive tracts. We performed a systems-level analysis of Rfx2-dependent gene expression, and we used functional gene networks to guide exploration of the resulting datasets. By coupling this approach to validation by in vivo imaging, this study provides important insights into the control of MCC gene expression (*Figure 8*, left) and into the molecular control of discrete MCC cell behaviors (*Figure 8*, right). Moreover, the work dramatically expands our understanding of the vertebrate RFX family transcription factors in ciliogenesis, in cilia function, and surprisingly, in cell movement.

### Rfx2 and the genetic network of multiciliated cell development and function

In addition to Rfx2, recent work has identified keys roles for the transcription factors Mcidas and Foxj1 in the development of MCCs (*You et al., 2004*; *Stubbs et al., 2008, 2012*). A key challenge going

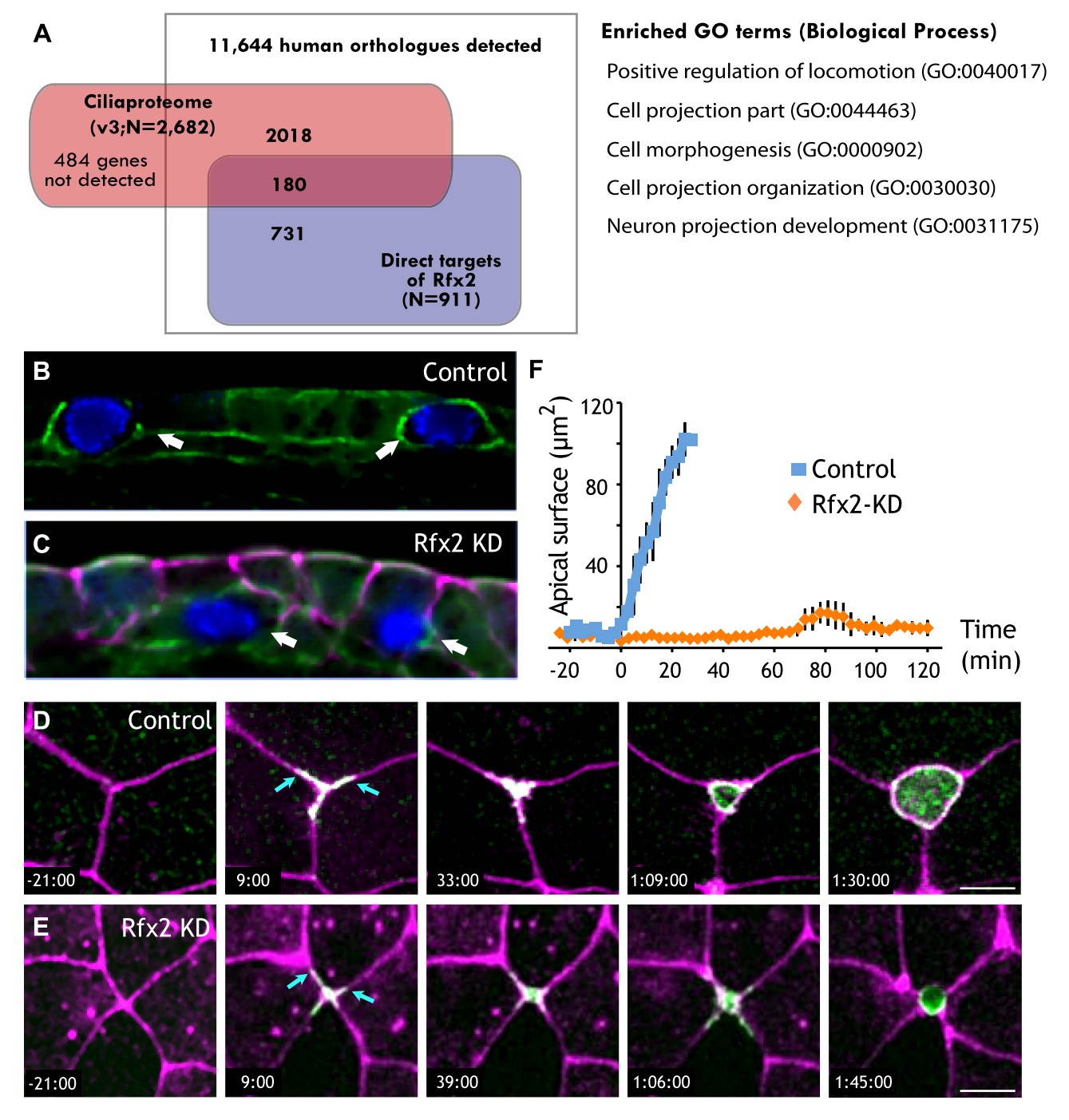

Figure 5. Rfx2 is essential for the insertion of nascent MCCs into the mucociliary epithelium. (A) Overlap of Rfx2 target genes and the 'cilia proteome' (as defined in **Gherman et al., 2006**; see 'Materials and methods'). Out of 911 direct target genes of Rfx2 identified in this study, only 20% of them (180 genes) are annotated as known cilia genes. Right panel represents the Gene Ontology terms significantly enriched among direct targets of Rfx2 (biological process category only; Benjamini corrected p<0.05) (B) A cross-sectional view of a control embryo labeled with ciliated cell marker (cyan). Apical surface is up. MCCs have inserted into the mucociliary epithelium (arrows). (C) A cross-sectional view of an Rfx2 morpholino-injected embryo. MCCs fail to insert into the overlying epithelium (arrows). To observe the insertion of MCCs into the overlying epithelium of control embryos (E) and Rfx2 morphants (D), a MCC-specific α-tubulin enhancer element driving expression of Utrophin-GFP was used. (D) Note the control MCC first exhibited a star-shaped morphology and cell protrusions probed into overlying cell–cell boundaries (arrows). The probing phase then ceased and apical surface expanded (**Videos 5 and 6**). (E) Protrusions of the MCC were observed, indicating the initial probing was qualitatively normal following Rfx2 knockdown. However, apical surface expansion was strongly inhibited in MCCs (**Video 7**). (F) Quantification of apical surface area of MCCs of control embryos and Rfx2 morphants.

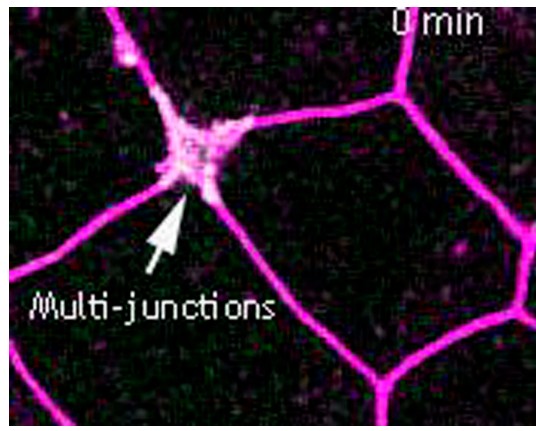

**Video 5**. The development of a control multiciliated cell. A control multiciliated cell expressing membrane-GFP is shown.

forward will be to understand how Rfx2 fits into this larger gene regulatory network, so we compared our set of direct Rfx2 targets with the downstream transcriptome of these other factors (*Figure 8*, left).

Mcidas has been implicated in the earliest stages of MCC specification (*Stubbs et al., 2012*), and accordingly, both Rfx2 and FoxJ1 are downstream of Mcidas, while Mcidas is not present in the target gene sets of the other two factors. Interestingly, our comparison suggests that many ciliary machines require the combined action of all or a subset of these three factors. For example, *ift80* is downstream of all three factors, while *Ift46* and *ift57* are present only in the FoxJ1 gene set and *ift140*, *ift172*, *ift81*, and *ift88* are targets only of Rfx2. Such combined action by Rfx factors and FoxJ1 is consistent with recent reports in mammalian airway epithelium (*Didon et al., 2013*) and in *Drosophila* (*Newton et al., 2012*). Nonetheless, our data also suggest that Rfx2 also plays essential roles in ciliogenesis that are independent of FoxJ1 or Mcidas, as many IFT genes and several ciliopathy-associated genes (*cep164*, *rpgripl1* and *cc2d2a)* are under the sole control of Rfx2 (*Figure 8*, left).

Moreover, we have discovered a role here for Rfx2 in apical surface assembly of nascent MCCs, and Foxj1 has no known role in this process. Accordingly, the MCC insertion genes *dag1* and *slit2* were present only in the Rfx2 target gene list and were not identified as downstream targets of FoxJ1 or Mcidas. Interestingly however, Rab11 is also required for MCC insertion (*Kim et al., 2012*), but was not present among our Rfx2 targets. Future studies will be needed to fully understand the gene regulatory network of MCCs, including how other essential transcription factors (e.g., Myb [*Tan et al., 2013*]), collaborate with the factors discussed here. Finally, it will now be of great interest to ask how this emerging gene regulatory network governing cilia structure and function in multiciliated cells compares with that governing the structure and function of primary, non-motile cilia in other cell types.

## From Rfx2-dependent gene expression to cell biological mechanism

Our analysis revealed that Rfx2 controls cilia structure and function by modulating expression of dozens of known genes, and this dataset serves as an important complement to genomic studies of RFX factors in invertebrates (*Efimenko et al., 2005*; *Laurencon et al., 2007*; *Newton et al., 2012*; *Phirke et al., 2011*; *Burghoorn et al., 2012*). However, a large proportion of genes, even in model animals, remain uncharacterized, and our Rfx2 target gene list contains dozens of genes with no known function. To overcome this hurdle, we used a functional gene network to identify important functional interactions among novel genes and known genes in the Rfx2 target gene set and we used in vivo imaging to test these interactions (*Figure 8*, right). This approach led us to discover an essential role for Ttc29 in the control of anterograde IFT and ciliogenesis and to begin delineating a functional hierarchy for the novel proteins Ribc2 and Nme5 in ciliary beating.

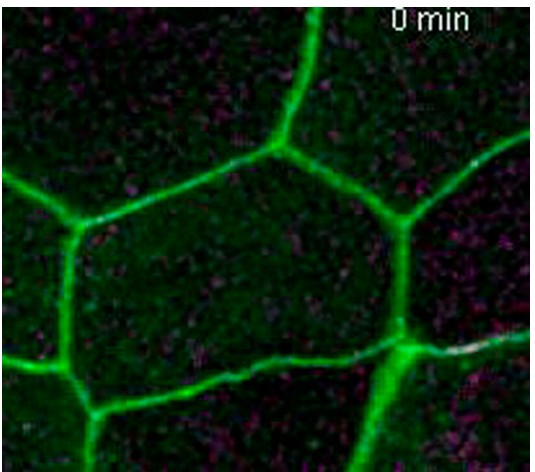

**Video 6**. The insertion of a control multiciliated cell into the overlying epithelium. A control multiciliated cell expressing Utrophin-RFP is shown. Note the control multiciliated cell first exhibited a probing phase and then an apical surface expansion phase.

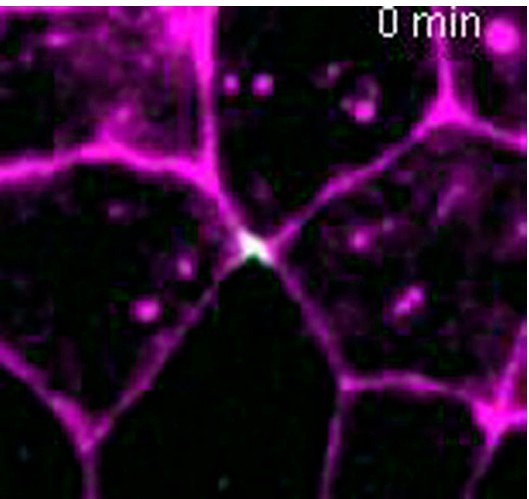

**Video 7**. The insertion of a Rfx2-knockdown multiciliated cell into the overlying epithelium. A Rfx2-knockdown multiciliated cell expressing Utrophin-GFP is shown. The initial probing was qualitatively normal following Rfx2 knockdown. However, apical surface expansion was strongly inhibited.

Moreover, guided by our exploration of gene networks, we found that the Rfx2 targets Tekt2 and Nme5 localize in novel proximodistally-restricted patterns along the axoneme, a result that is of interest because precise proximodistal positioning of specific dynein arm proteins along motile axonemes is central to normal cilia beating. Indeed, though defective proximodistal pattern in axonemes is associated with human primary ciliary dyskinesia (PCD) (*Fliegauf et al., 2005*; *Panizzi et al., 2012*), we know essentially nothing of how these patterns are controlled. Thus, by exploiting our genomic data to guide cell biological inquiry, this study has provided a foundation for future exploration of this problem.

## Cilia-independent roles for Rfx2 and apical surface assembly in nascent MCCs

The role of RFX factors in ciliogenesis has been a key focus of recent research, so it is notable that this function is not thought to be ancestral for RFX proteins (*Piasecki et al., 2010*). Indeed, the yeast RFX orthologue, Sak1, controls cell cycle exit (*Wu and McLeod, 1995*) and even Rfx factors commonly associated with ciliogenesis (e.g., Daf-19, Rfx3) perform cilia-independent functions (*Senti and Swoboda, 2008*; *Ait-Lounis et al., 2010*). Accordingly, we found that Rfx2 also controls the assembly of the apical surface of nascent MCCs as they insert into the epithelium. Rfx2 effects this process via the known regulator *dag1* and through *slit2*, which we show here to be essential for MCC apical surface expansion. These results are of particular interest both because Dag1 and Slit/Robo signaling also collaborate during axon growth

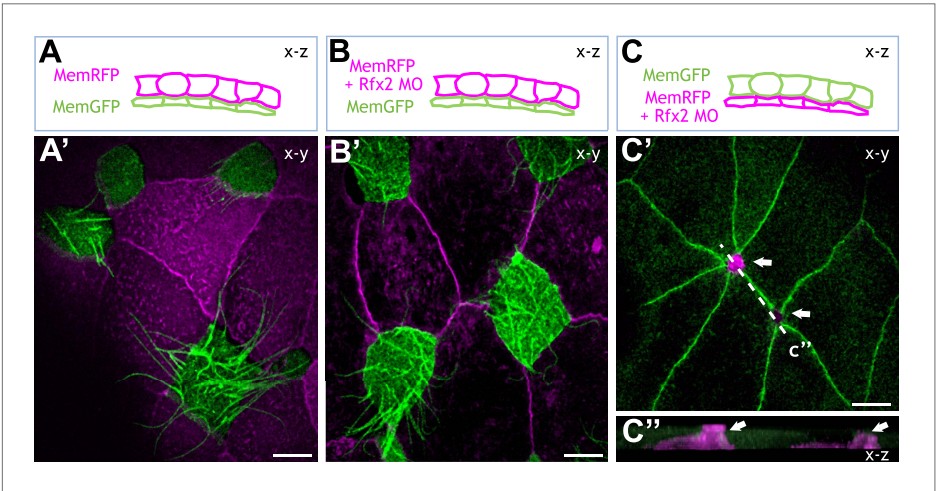

**Figure 6**. Rfx2 acts cell-autonomously to control insertion of nascent MCCs into the overlying epithelium. (**A**)–(**C**) Illustration of the transplantation experiments. The superficial layer from either control (**A**) (**A'**) or Rfx2 knockdown embryos (**B**) (**B'**) was transplanted to the control host embryos. (**C**) (**C'**) the superficial layer from control embryos was transplanted to the Rfx2 knockdown embryos. At stage 26, MCCs derived from the control host have intercalated into the outer layer transplanted from either control (**A'**) or Rfx2 knockdown embryos (**B'**). (**C'**) MCCs, in which Rfx2 was knocked down, failed to insert properly into control outer epithelium. (**C''**) A z-view of two MCCs in (**C'**).

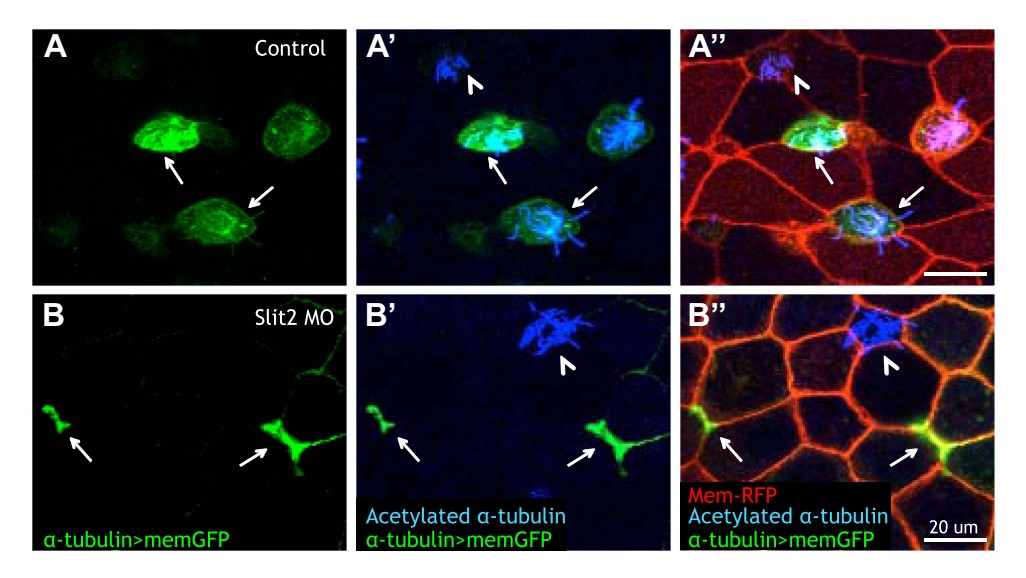

**Figure 7**. Slit2 is required for MCC insertion into the overlying epithelium. (**A**) A control embryo injected with membrane-RFP and α-tubulin > membrane-GFP to label MCCs. Injected embryos were then fixed and stained with RFP, GFP, and α-acetylated tubulin. (**B**) Slit2 morpholino was injected with membrane-RFP and α-tubulin > membrane-GFP. Note that MCCs fail to insert into the mucociliary epithelium following Slit2 knockdown (arrows). In addition, MCCs (arrowheads) with no Slit2 morpholino insert into the superficial layer containing Slit2 morpholino. These data indicate Slit2 controls MCCs in a cell-autonomous fashion.

cone guidance (*Andrews et al., 2006*; *Wright et al., 2012*) and because Rfx3 mutant mice display axon guidance defects (*Benadiba et al., 2012*). These data may also shed light on the mechanism of Rfx action in synapse morphogenesis in *C. elegans* (*Senti and Swoboda, 2008*), as the neuron projection morphogenesis GO term was enriched in our Rfx2 target gene set, reflecting that mediators of synapse morphogenesis, such as PTPδ and Netrin3, were present in the target set (*Supplementary file 1C*) (*Takahashi et al., 2012*). Finally, both the genomic studies and our in vivo imaging of MCC insertion provide an important foundation for future work, because mammalian MCCs—like the *Xenopus* MCCs studied here—arise from basally-positioned precursor cells (*Evans and Moller, 1991*; *Rock et al., 2010*).

In conclusion, this study has revealed a central role for Rfx2 at the nexus of cell movement, ciliogenesis, and cilia motility (*Figure 8*). The work sheds new light on Rfx2 protein functions specifically and also on the transcriptional control of cell behavior and organelle biogenesis generally. Our combined approach of systems biology, computational biology, and in vivo cell biology provides a generalizable paradigm for exploiting genomic data to advance our understanding of cell biological processes.

## Materials and methods

### Morpholino and RNA injection
Capped mRNA was synthesized using mMESSAGE mMACHINE (Ambion, Austin, TX). mRNA and antisense morpholino were injected into ventral blastomeres at the 4-cell stage to target the epidermis (*Moody, 1987*). Embryos were incubated until appropriate stages and were fixed in MEMFA (*Davidson and Wallingford, 2005*). The embryos were embedded in 2% agarose for thick (250–300 micron) sections, which were cut with a Vibratome series 1000 (*Davidson and Wallingford, 2005*). Morpholino sequence and the working concentration:
    Rfx2 morpholino: AATTCTGCATACTGGTTTCTCCGTC, 12 ng
    Ttc29 morpholino: GTGCACTCATTCTCTTCAAGTTTGC, 40 ng
    Ribc2 morpholino: CGATAGGCAGATCCAGTCGGTACAT, 21 ng
    Slit2 morpholino: TTCAGGTCTCTGGGAAAACAGGAAC, 10 ng

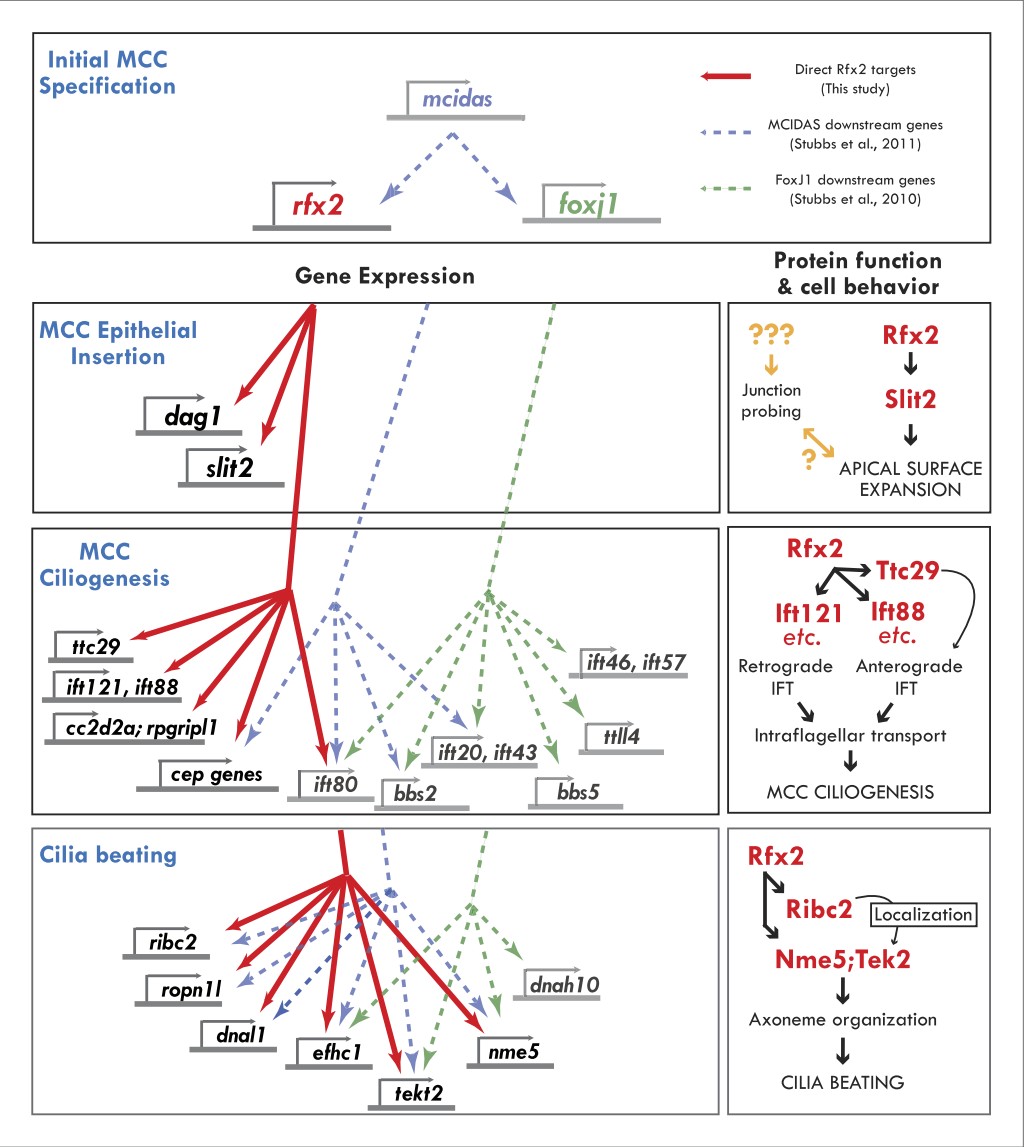

**Figure 8**. Genomic control of cellular functions in developing multiciliated cells. Left panels ('Gene Expression') illustrate genetic controls as reported in **Figure 2**, **Figure 2—figure supplements 2–4**, and **Supplementary file 1A–C**. Solid red lines in this figure indicate direct control of transcription by Rfx2 (e.g., intersect of RNAseq and ChIPseq data as outlined in **Figure 2**). Previous genomic analyses of Mcidas and Foxj1 in Xenopus MCCs did not include ChIPSeq data, so the dashed blue and green lines shown here indicate only that a gene's transcription was upregulated in response to overexpression of *mcidas* or *foxj1*, respectively as per (**Stubbs et al., 2008**; **Stubbs et al., 2012**). Right panels ('protein function and cell behavior') illustrate mechanistic insights found here for MCCs by live imaging studies (**Figures 3–7**, **Figure 4—figure supplements 2 and 3**).

## Draft genome and annotation of *Xenopus laevis*

We used draft version (version 6, JGIv6) of the *Xenopus laevis* genome for analyzing genomic and ChIP-seq datasets, obtained from the International *Xenopus laevis* genome project consortium and now available from the XenBase FTP site (ftp://ftp.xenbase.org). We selected scaffolds longer than 10,000 bases for further analysis, for a total of 8426 scaffolds used in this analysis. Because gene models for *X. laevis* are not yet finalized, we employed interim gene models for the analysis described here, using the released transcriptome-derived gene models ('Oktoberfest' version), also provided by the International *Xenopus laevis* genome project. Sequences and detailed descriptions of the gene

model construction pipeline are available at the project website (http://www.marcottelab.org/index.php/Xenopus_Genome_Project).

## Differential expression following Rfx2 knockdown as measured by RNA-seq

Total RNA was collected from 100 animal caps each at stage 20 from control embryos and from Rfx2 morphants. After poly-(A)-capture, we prepared sequencing libraries using the standard manufacturer's non-strand specific Ilumina RNA-seq protocol, and sequenced paired-end 2 × 50 bp reads by Ilumina HiSeq 2000. Reads were mapped to the longest transcripts for each of the 'Oktoberfest' gene models using the bowtie mapper (version 0.12.7; '-a -v 2' options were applied) (*Langmead et al., 2009*). We then estimated normalized gene expression values for each sample by calculating RPKMs (Reads Per Kilobase per Million mapped reads). It should be noted that one of the two Rfx2 MO replicates (M2) had significantly fewer reads (30 M read pairs) than the other (72 M read pairs). However, measured expression levels (RPKMs) correlated well between the replicates (*Figure 2—figure supplement 2*), so we retained both for subsequent analyses. Using the edgeR package (*Robinson et al., 2010*), we identified significant changes in RNA abundances between control and Rfx2 knockdowns, requiring greater than twofold abundance changes and a false discovery rate (FDR) less than 5%. Out of 11,644 genes tested, we identified 2750 genes significantly differentially expressed following Rfx2 knockdown.

## Rfx2 chromatin-immunoprecipitation and sequencing (ChIP-seq)

ChIP-seq was performed as described previously (*Kim et al., 2011*) to identify direct chromosomal binding sites of Rfx2. Briefly, ChIP-seq assays were performed with stage 20 *X. laevis* embryos, injected with in vitro transcribed mRNA coding for either GFP-Rfx2 or GFP alone. Samples were crosslinked with 1% formaldehyde for 1 hr and the reaction stopped by adding glycine (to 125 mM) for 10 min. The embryos were rinsed with PBS and resuspended in lysis buffer with protease inhibitor cocktail (Roche). Chromatin was sonicated to an average size of 200–600 base pair using a Branson 450 Sonifier, then immunoprecipitated using protein G magnetic beads (Invitrogen) coupled to 5 µg α-GFP antibody (ab290) at 4°C overnight. Magnetic beads were washed, the bound chromatin eluted, and crosslinks reversed. ChIP DNA was extracted with phenol-chloroform and purified with a QIAquick PCR Purification Kit (Qiagen). ChIP-seq libraries were prepared according to the standard manufacturer's Illumina sequencing protocol and sequenced by Illumina HiSeq.

Reads were mapped to the *Xenopus laevis* draft genome (version 6) using bowtie ('-m 1 -n 2' options were applied) and peaks identified using MACS (version 1.4.2) (*Zhang et al., 2008*) with default options. Out of 29,448 peaks identified, we selected 6646 peaks for further study that exhibited either a FDR <5% or a fold-enrichment > 20. Genes were associated with significant ChIP-seq peaks based on proximity in the draft genome, requiring genes (specifically, the longest transcript for each gene model as mapped to the draft genome sequence) to lie within 10,000 bases from an identified peak. Out of the 6646 significant ChIP-seq peaks, 3465 peaks could be assigned to nearby genes and 911 of those putative target genes also showed significantly different gene expression after Rfx2 knockdown. We focused on these differentially expressed, directly bound genes for subsequent analyses (*Figure 2—source data 1*).

## Analysis of RFX2 targets using the HumanNet gene network

To better understand the molecular mechanisms of genes regulated by RFX2, we analyzed the clustering of RFX2 target genes using a human functional gene network ('HumanNet') (*Lee et al., 2011*). To increase potential coverage, we considered *X. laevis* genes that were either marked by RFX2 binding sites in the ChIP-seq data or that were significantly differentially expressed following RFX2. For the human orthologs of these genes (as defined by the International *Xenopus laevis* genome project using phylogenetic analyses of gene models during the course of annotating *X. laevis* genes), we extracted HumanNet gene–gene linkage information and associated confidence scores (log likelihood scores; LLS). The resulting network contained 4609 genes with 52,714 functional linkages, and served as the basis for later analyses.

Due to an ancestral genome duplication along the *X. laevis* lineage, many human genes have (typically) two *X. laevis* orthologs, generally referred to as 'homeologs' or 'alloalleles'. For the purposes of calculating gene networks among RFX2 target genes, we transferred evidence from either of the alloalleles to the single orthologous human gene. For example, in the case of having one human

gene orthologous to two frog genes, if one of the frog homologs was identified as a direct target of RFX2 and the other was not, we considered the human gene as a direct target of RFX2 for the purposes of reconstructing the network. To identify functional modules regulated by RFX2 in unbiased way, we clustered this network using the clusterONE algorithm (*Nepusz et al., 2012*) available in Cytoscape (version 2.8.3) (*Shannon et al., 2003*; *Cline et al., 2007*), considering linkage confidence scores during the clustering. All network information is available at the following URL: http://www.marcottelab.org/index.php/ChungKwon2013_RFX2

## Comparison of Rfx2 targets to the cilia proteome

For the comparison of Rfx2 targets to ciliary proteins in *Figure 5A*, we used a compiled list of ciliary proteins drawn from several studies using a combination of proteomics and comparative genomics (*Gherman et al., 2006*). The protein set was downloaded from this website, http://v3.ciliaproteome. org/cgi-bin/protein_browser.php, then converted to EnsEMBL gene IDs using BioMart (version 63).

## In situ hybridization

In situ hybridization was performed as described previously (*Sive et al., 2000*). Bright field and low magnification fluorescence images were captured on a fluorescent stereomicroscope, Leica MZ16FA.

## Immunohistochemistry

Embryos were fixed in MEMFA for 1 hr followed by washing in PTW (PBS+0.1% Tween 20) for 30 min (3 × 10 min). Embryos were then blocked in fetal bovine serum (FBS) solution (TBS containing 10% FBS and 5% DMSO) for 1 hr at room temperature. Embryos were then incubated with the following primary antibodies at 4°C overnight: monoclonal anti-α-tubulin antibody (1:500 dilutions, clone DM1A, Sigma), mouse anti-acetylated-α-tubulin (1:500, clone 6-11B-1, Sigma), chicken anti-GFP antibody (1:500, ab13970), and rabbit anti-RFP (1:500, ab62341). After primary antibody incubation, all samples were washed with TBST (TBS+0.1% Triton X-100) for 5 hr (1 × 5 hr). Primary antibodies were detected with Alexa Fluor 488 goat anti-mouse IgG (1:500, Molecular Probes), Alexa-555 goat anti-rabbit IgG (1:500, Molecular Probes), and Alexa-488 goat anti-chicken IgG (1:500, Molecular Probes). After secondary antibody incubation, all samples were washed with TBST for 5 hr.

Embryos prepared for confocal imaging as described (*Wallingford, 2010*). Images were obtained using Zeiss LSM5 Pascal and Zeiss LSM700 confocal microscope. Cilia lengths were measured with Fiji software. Images used throughout this paper have been enhanced using the Unsharp Mask filter in Adobe Photoshop.

## Confocal imaging of live embryos

High-speed confocal imaging was performed by time-lapse collection of single optical section at a frame rate of 370fps using a Zeiss LSM 5LIVE microscope. Images were collected from living embryos expressing membrane-GFP driven by MCC-specific promoter and from embryos injected with Ribc2 morpholino.

For filming the MCC intercalation, living embryos of either control or Rfx2 morpholino-injected animals were put on a round cover glass in custom machined dishes (*Kieserman et al., 2010*). Embryos were gently pushed down by a small piece of cover glass. Images were collected every 3–5 min and were then processed into a time-lapse movie using Fiji software.

For IFT imaging, embryos expressing GFP-IFT20 alone or with Ttc29 MO were mounted flank down in 0.8% LMP agarose. Single confocal slices were collected at ~2 fps using an LSM 5LIVE confocal microscope, as previously described in *Brooks and Wallingford (2012)*.

## Cell-autonomy/transplantation assays

Embryos were injected ventrally at 4-cell stage with mRNA encoding either membrane-GFP or membrane-RFP. Rfx2 morpholino was injected together with membrane-RFP. At stage 10, a fine hair was used to peel off the outer layer from a region of the ectoderm of a donor embryo. This outer layer peel was then transferred onto a host embryo after removing a similar patch of outer cells. To help the healing process, a small piece of glass coverslip with clay feet was used to press down embryos. Transplantations were performed in Danilchick's Solution for Amy (DFA) + 0.1% BSA. After healing, embryos were then transferred back to 1/3 MMR (*Stubbs et al., 2006*).

## Acknowledgements

We thank The International *Xenopus laevis* Genome Project Consortium, especially the Harland, Rokhsar, and Taira labs, as well as all data contributors to the consortium. We also thank Scott Hunicke-Smith and The Genome Sequencing and Analysis Facility (GSAF) at UT Austin for RNA-seq and the Texas Advanced Computing Center (TACC) for supercomputer use. JBW is an Early Career Scientist of the Howard Hughes Medical Institute.

## Additional information

### Funding

| Funder | Grant reference number | Author |
| --- | --- | --- |
| National Institute of General Medical Sciences | | Julie C Baker, Edward M Marcotte, John B Wallingford |
| Howard Hughes Medical Institute | | John B Wallingford |
| Cancer Prevention Research Institute of Texas | | Edward M Marcotte |
| National Science Foundation | | Edward M Marcotte |
| The US Army | 58343-MA | Edward M Marcotte |
| The Welch Foundation | F1515 | Edward M Marcotte |
| National Heart, Lung, and Blood Institute | | John B Wallingford |

The funders had no role in study design, data collection and interpretation, or the decision to submit the work for publication.

### Author contributions

M-IC, TK, Conception and design, Acquisition of data, Analysis and interpretation of data, Drafting or revising the article; FT, ERB, Conception and design, Acquisition of data, Analysis and interpretation of data; RG, JCB, Conception and design, Acquisition of data, Drafting or revising the article; MM, Acquisition of data, Analysis and interpretation of data; EMM, JBW, Conception and design, Analysis and interpretation of data, Drafting or revising the article

### Ethics

Animal experimentation: All animal experiments were approved by the IACUC of the University of Texas at Austin, Protocol# AUP-2012-00156.

## Additional files

### Supplementary files

• Supplementary file 1. (A) Rfx2 target genes controlling ciliogenesis. Target genes are listed by gene ID. Column 2 displays log2-transformed fold-change for each genes between control and Rfx2 morphants (green for down-regulated, red for up-regulated). Column 3 displays the adjusted *p*-value for differential expression after multiple hypothesis testing correction. Columns 4 and 5 display the numbers of RNA-seq reads (not normalized) for control samples in each of two biological replicates; columns 6 and 7 show the number of RNA-seq reads (not normalized) for Rfx2 morphant samples in each of two biological replicates. Column 8 provides the peak fold enrichment for Rfx2 binding sites relative to the GFP-only control ChIP; Column 9 is the false discovery rate (%) of Rfx2 ChIP-seq peak calls. (B) Rfx2 target genes controlling cilia beating. Target genes are listed by gene ID. Columns are as per *Supplementary file 1A*. (C) Rfx2 target genes related to Slit/Robo signaling and neuronal migration/morphogenesis. Target genes are listed by gene ID. Columns are as per *Supplementary file 1A*.

## Major dataset

The following dataset was generated:

| Author(s) | Year | Dataset title | Dataset ID and/or URL | Database, license, and accessibility information |
|---|---|---|---|---|
| Chung M, Kwon T, Gupta R, Baker JC, Marcotte EM, Wallingford JB | 2013 | Coordinated genomic control of ciliogenesis and cell movement | GSE50593; http://www.ncbi.nlm.nih.gov/geo/query/acc.cgi?acc=GSE50593 | Publicly available at GEO (http://www.ncbi.nlm.nih.gov/geo/). |

The following previously published datasets were used:

| Author(s) | Year | Dataset title | Dataset ID and/or URL | Database, license, and accessibility information |
|---|---|---|---|---|
| *Xenopus laevis* genome project consortium | 2012 | *Xenopus laevis* genome (JGI version 6.0) | ftp://ftp.xenbase.org/pub/Genomics/JGI/Xenla6.0/ | Publicly available at XenBase (http://www.xenbase.org/), the Xenopus model organism database. |
| *Xenopus laevis* genome project consortium | 2012 | *Xenopus laevis* gene model (Oktoberfest version) | ftp://xenbaseturbofrog.org/sequence_information/UTA/ | Publicly available at XenBase (http://www.xenbase.org/), the Xenopus model organism database. |
| Lee I, Blom UM, Wang PI, Shim JE, Marcotte EM | 2011 | HumanNet - probabilistic functional network of human | http://www.functionalnet.org/humannet/HumanNet.v1.join.txt | Publicly available on the project website (http://www.functionalnet.org/humannet/). |

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
