## [Decision Letter]

Thank you for sending your work entitled “Coordinated genomic control of ciliogenesis and cell movement by RFX2” for consideration at *eLife*. Your article has been favorably evaluated by a Senior editor and 3 reviewers.

The Reviewing editor, Marianne Bronner, and the reviewers discussed their comments before we reached this decision, and the Reviewing editor has assembled the following comments to help you prepare a revised submission.

The paper entitled by Chung et al. describes the role of the transcription factor RFX in driving expression of genes involved in diverse cellular functions in the multiciliated cells of *Xenopus epithelium*. Specifically, they perform both RNA-seq (wild type versus morphant) and DNA Chip-Seq to identify genes directly bound and upregulated by RFX. They then use these data to perform computational analysis that assigns unknown genes into possible cellular processes including ciliogenesis, cilia beating, and cell migration. Finally with a subset of novel targets they test function using gene specific morphants. They identify several uncharacterized proteins that are involved in IFT, in cilia beating and in migration.

This manuscript contains an impressive amount of data that deal with an important biological issue. In addition, it describes a novel mode for identifying potentially important proteins. The authors do a nice job of putting their data into the broader context of what is known in the field. That said, there some additions, detailed below, that are required to render this paper appropriate for publication in *eLife*.

1) It is essential that the authors provide a list of all of the 911 genes identified in their screen in the form of an excel sheet. This is critical so that the community can compare and contrast the quality and usefulness of the data. In this way, this “screen paper” would provide a truly useful and important data set for the community.

2) The authors should provide a complete description of the technique and some more detailed analysis of the gene set, for example using gene ontology. While knocking down 911 genes is wholly impractical and clearly beyond the scope of the present paper, the use of gene ontology groupings may identify other potential modules and provide new hypotheses into the roles of Rfx2 in early epithelial development.

3) In their diagram, the authors place Mcidas above RFX due to the fact that RFX and FOXJ1 came up from the Mcidas screen, which makes sense. However, RFX is important for multiple types of cilia, including primary, whereas Mcidas is not required for primary cilia. This causes confusion over the hierarchy of these factors and should be discussed on a cell type specific basis.

---

## [Author Response]

*1) It is essential that the authors provide a list of all of the 911 genes identified in their screen in the form of an excel sheet. This is critical so that the community can compare and contrast the quality and usefulness of the data. In this way, this “screen paper” would provide a truly useful and important data set for the community*.

This is an excellent point and we regret the oversight here. We hasten to note that the entire dataset was submitted to the NCBI, but we also acknowledge that a table included in the paper will be far more useful. This table has been added as Figure 2–source data 1.

*2) The authors should provide a complete description of the technique and some more detailed analysis of the gene set, for example using gene ontology. While knocking down 911 genes is wholly impractical and clearly beyond the scope of the present paper, the use of gene ontology groupings may identify other potential modules and provide new hypotheses into the roles of Rfx2 in early epithelial development*.

We have included a detailed description of our techniques in the Materials and methods section of the paper. We have also now added a supplemental table of enriched GO terms (Figure 2–source data 2). Satisfyingly, this analysis finds that cilia-related and cell locomotion/cell projection GO terms are the most enriched. Moreover, we also find a neuron projection category that is enriched, in line with the discussion of our paper. Finally, we find a fourth category (nucleotide/nucleoside related terms) that we had not previously appreciated. This information has now been presented in the Results and Discussion sections.

*3) In their diagram, the authors place Mcidas above RFX due to the fact that RFX and FOXJ1 came up from the Mcidas screen, which makes sense. However, RFX is important for multiple types of cilia, including primary, whereas Mcidas is not required for primary cilia. This causes confusion over the hierarchy of these factors and should be discussed on a cell type specific basis*.

We regret that we were unclear on this point. It is true that Rfx2 is required for primary ciliogenesis. However, our RNAseq and ChIPseq were performed on multi-ciliated cells, so we do not yet know the extent to which these findings will apply to primary cilia.

We have therefore clarified the Figure, the figure legend, and the manuscript text to emphasize that our findings apply specifically to multi-ciliated cells.